# One-dimensional purely Lee-Huang-Yang fluids dominated by quantum fluctuations in two-component Bose-Einstein condensates

Xiuye Liu[1,2] and Jianhua Zeng[1,2,*]

[1]*State Key Laboratory of Transient Optics and Photonics,*
*Xi'an Institute of Optics and Precision Mechanics of Chinese Academy of Sciences, Xi'an 710119, China*
[2]*University of Chinese Academy of Sciences, Beijing 100049, China*

Lee-Huang-Yang (LHY) fluids are an exotic quantum matter dominated purely by quantum fluctuations. Recently, the three-dimensional LHY fluids were observed in ultracold atoms experiments, while their low-dimensional counterparts have not been well known. Herein, based on the Gross-Pitaevskii equation of one-dimensional LHY quantum fluids in two-component Bose-Einstein condensates, we reveal analytically and numerically the formation, properties, and dynamics of matter-wave structures therein. Considering a harmonic trap, approximate analytical results are obtained based on variational approximation, and higher-order nonlinear localized modes with nonzero nodes are constructed numerically. Stability regions of all the LHY nonlinear localized modes are identified by linear-stability analysis and direct perturbed numerical simulations. Movements and oscillations of single localized mode, and collisions between two modes, under the influence of different initial kicks are also studied in dynamical evolutions. The predicted results are available to quantum-gas experiments, providing a new insight into LHY physics in low-dimensional settings.

**keywords**: Bose-Einstein condensates, Cold atoms, Lee-Huang-Yang fluids, Quantum fluctuations

## I. INTRODUCTION

Realizations of Bose-Einstein condensates (BECs) in 1995 for rubidium-87 [1], sodium-23 [2] and Lithium-7 [3] atoms had heralded the landmark of modern physics, then studies of coherent matter waves sparked renewed attention of scientists from various fields [4–8], evidenced by numerous theoretical predictions and the subsequent experimental observations of novel quantum states of matter driven by BECs, which include degenerate quantum Fermi gases [9], Tonks-Girardeau gases with impenetrable (hard-core) bosons [10], spin-orbit-coupled BEC [11, 12], and quantum droplets [13–22]. Particularly, quantum droplets [13, 14] are described by mean-field theory as that for BECs and by beyond mean-field contribution— Lee-Huang-Yang (LHY) correction induced by quantum fluctuations [23, 24]. Competing interplays of atom-atom interactions between mean-field term and LHY correction have also measured in dipolar quantum gases (quantum droplets in single-component BEC) [25].

Quantum fluctuations as many-body effects are critical in controlling ultracold atomic gases where beyond-mean-field correction is relevant. LHY physics describing for weak quantum fluctuations has been demonstrated in ultracold atomic gases experiments, such as BECs loaded in optical lattices (which decrease kinetic energy of a single particle and increase atom-atom interactions) [26, 27], quantum phase transitions from superfluid to Mott insulators [28–32], quantum criticality [33] and the Tonomaga-Luttinger liquids [34]. Another setting is provided by Feshbach resonance management that can tune the sign and value of interatomic interactions by controlling the associated scattering length [35–

38], according to initial experimental confirmations of the frequency shifts in collective oscillations [39] and density profiles [40, 41], both deviated from mean-field theory, in the fermionic condensates. In strongly interacting bosonic gases, effects of tunable interaction strengths beyond the mean-field regime were also experimentally observed in excitation spectrum of ultracold rubidium-85 atoms [42] and in the state equation of lithium-7 condensates [43]. In spinor BECs, quantum fluctuations completely lift the accidental degeneracy in ground-state manifold [44, 45]and greatly increase the quantum mass (acquisition) of quasiparticles [46]. Additonally, momentum-resolved observation of quantum and thermal depletion was reported in metastable He [47] and the density of total quantum depletion in potassium-39 [48].

In a Bose-Bose mixture without any external trapping, the repulsive LHY correction accounted for quantum fluctuations could neutralize attractive mean-field term and therefore stabilizes the otherwise collapsing atomic system against the onset of self-focusing critical (supercritical) collapse, forming a balanced state called quantum droplets [13, 14]. In similar systems, where interspecies attraction $g_{12}$ balances intraspecies repulsive $(g_{11}, g_{22})$, $g_{12} = -\sqrt{g_{11}g_{22}}$, and atom number fulfills $N_2 = \sqrt{g_{11}/g_{22}}N_1$, the mean-field terms would cancel and lead to a new quantum matter called LHY quantum fluids governed merely by quantum fluctuations [49]. Note, importantly, that such a new three-dimensional (3D) quantum state has been observed very recently in a potassium-39 spin mixture confined in a spherical trap potential [50]. To our knowledge, physics of low-dimensional purely LHY fluids remains elusive.

In this work we obtain a theoretical framework for one-dimensional (1D) LHY fluids in two-component BECs, and explore analytically and numerically the formation, properties, and dynamics of localized wave structures thereof. The framework is based on Gross-Pitaevskii equation with focusing quadratic nonlinearity, radically different from its 3D counterpart with defocusing quadruple nonlinearity, offering a new model for investigating bright matter-wave structures.

* zengjh@opt.ac.cn

We construct numerically approximate solutions of fundamental modes using variational approximation and higher-order nonlinear modes with nonzero nodes $\kappa = 1$ and $2$ in the presence of harmonic trap, report motion and oscillations of single mode and collisions between the two modes under different incident momenta in dynamical evolutions.

## II. THE MODEL

The energy functional for two-component BECs, including the mean-field term and weakly LHY contribution (quantum fluctuations), reads [13, 14]

$$E = \frac{1}{2}\sum_{ij} g_{ij}n_i n_j + \frac{1}{2}\sum_{\pm,|\mathbf{k}|<\kappa}\left[E_{\pm}(k) - \frac{k^2}{2} - c_{\pm}^2\right], \quad (1)$$

here $k$ is the momenta and $n$ the density of BEC, and $E_{\pm}(k) = \sqrt{c_{\pm}^2 k^2 + k^4/4}$ being the Bogoliubov modes at sound velocities $c_{\pm}$ given by

$$c_{\pm}^2 = \frac{g_1 n_1 + g_2 n_2 \pm \sqrt{(g_1 n_1 - g_2 n_2)^2 + 4g_{12}^2 n_1 n_2}}{2}, \quad (2)$$

The momentum integration of Eq. (1) results in 1D energy functional [14]

$$E_{1D} = \frac{1}{2}\sum_{ij} g_{ij}n_i n_j - \frac{2}{3\pi}\sum_{\pm} c_{\pm}^3. \quad (3)$$

Substituting $c_{\pm}$ form Eq. (2) into above equation, leads to

$$\begin{aligned}E_{1D} =& \frac{\left(\sqrt{g_1}n_1 - \sqrt{g_2}n_2\right)^2}{2} - \frac{2}{3\pi}\left(g_1 n_1 + g_2 n_2\right)^{\frac{3}{2}} \\ &- \frac{g\left(g_{12} + \sqrt{g_1 g_2}\right)}{(g_1 + g_2)^2}\left(\sqrt{g_1}n_1 + \sqrt{g_2}n_2\right)^2.\end{aligned} \quad (4)$$

As required by LHY fluids [49] in a homonuclear mixture where the mean-field energy vanishes: $g_{12} = -\sqrt{g_1 g_2}$, $g = g_1 = g_2$, $n = n_1 = n_2$, then the energy functional, dominated merely by quantum fluctuations, becomes

$$E_{1D} = \frac{-4\sqrt{2}\,(gn)^{3/2}}{3\pi}. \quad (5)$$

Note that focusing (attractive) nonlinearity is the signature of 1D LHY fluids, in contrast to their 3D counterparts [49] that feature defocusing (repulsive) nonlinearity.

The Gross-Pitaevskii equation for the 1D LHY fluids describing by wave function $\Psi$ reads

$$i\hbar\frac{\partial\Psi}{\partial t} = -\frac{\hbar^2}{2m}\frac{\partial^2\Psi}{\partial x^2} - \frac{\sqrt{2m}}{\pi\hbar}g^{3/2}|\Psi|\Psi, \quad (6)$$

here $m$ is the atomic mass and Planck constant $\hbar$. With characteristic units $x = \frac{\pi\hbar^2}{\sqrt{2}mg^{2/3}}$, $t = \frac{\pi^2\hbar^3}{2mg^3}$, $\psi = \frac{\sqrt{2m}}{\pi\hbar}g^{\frac{3}{2}}\Psi$, we can get the normalized equation of motion [20]

$$i\frac{\partial\psi}{\partial t} = -\frac{1}{2}\frac{\partial^2\psi}{\partial x^2} + V(x)\psi - |\psi|\psi. \quad (7)$$

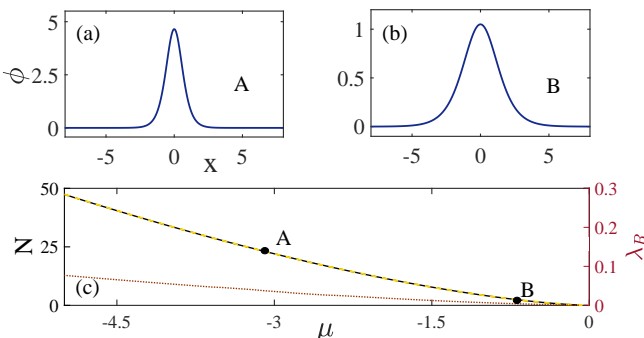

FIG. 1. Profiles, number of atoms ($N$) and linear-stability eigenvalues versus chemical potential $\mu$ for 1D LHY fluids in the absence of harmonic trap. Profiles of 1D LHY fluids at: (a) $\mu = -3.1$, $N = 23.1$; (b) $\mu = -0.7$, $N = 2.5$. (c) Dependence $N(\mu)$ (blue line represents calculated values, yellow dashed line represents analytical result given by Eq. (11) ) and maximal real values of eigenvalues $\lambda_R$ vs $\mu$ (red dashed line).

Where includes an external harmonic trap $V(x) = V_0 x^2$ with strength $V_0 = \frac{1}{2}\omega_x$ (frequency $\omega_x$).

Stationary solutions of Eq. (7) at chemical potential $\mu$ yield $\psi(x,t) = \phi(x)e^{-i\mu t}$, leading to stationary equation

$$\mu\phi = -\frac{1}{2}\frac{\partial^2\phi}{\partial x^2} + V(x)\phi - |\phi|\phi. \quad (8)$$

Based on the previous works for 1D quantum droplets in quadratic-cubic model in [14, 25, 51], discarding the cubic term in where can lead to an exact solution for our model Eq. (8) in the absence of harmonic trap,

$$\phi_{\text{exact}}(x) = \frac{-3\mu}{1 + \cosh\left(\sqrt{-2\mu}x\right)}. \quad (9)$$

Keep in mind that the number of atoms (norm) is defined as

$$N \equiv \int_{-\infty}^{\infty}|\phi(x)|^2\,dx, \quad (10)$$

Then the corresponding norm of the exact solution Eq. (9) is given by

$$N_{exact} = \frac{3\sqrt{2}\mu^2}{\sqrt{-\mu}}. \quad (11)$$

Stability property of localized modes for Eq. (7) is measured by linear-stability analysis. To this, we perturb the solutions as $\psi = [\phi(x) + p_+(x)e^{\lambda t} + p_-^*(x)e^{\lambda^* t}]e^{-i\mu t}$, with unperturbed mode $\phi$, tiny perturbations $p_+$ and $p_-^*$ at eigenvalue $\lambda$, and derive the stability equations

$$i\lambda p_{\pm} = \mp\frac{1}{2}\frac{\partial^2 p_{\pm}}{\partial x^2}\mp\mu p_{\pm}\mp\frac{1}{2}\left(3|\phi|\,p_{\pm} + \frac{\phi^2}{|\phi|}p_{\mp}\right)\pm V(x)p_{\pm}. \quad (12)$$

Where the localized modes $\phi$ are found from Eq. (8) by Newton-Rapson iteration, then such modes' stability is measured by solving the eigenvalue equations [Eqs. (12)], it is

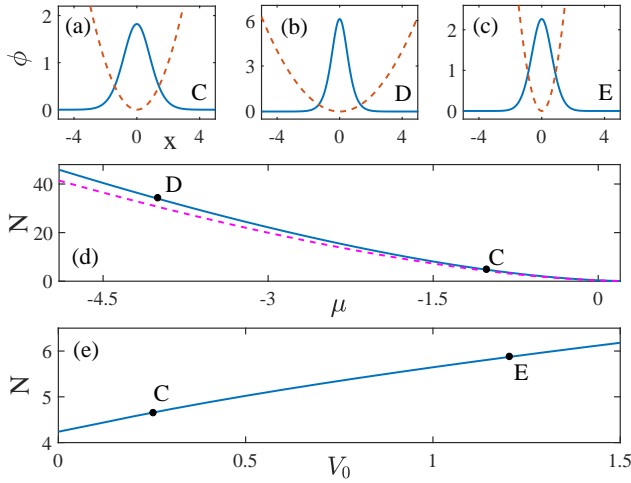

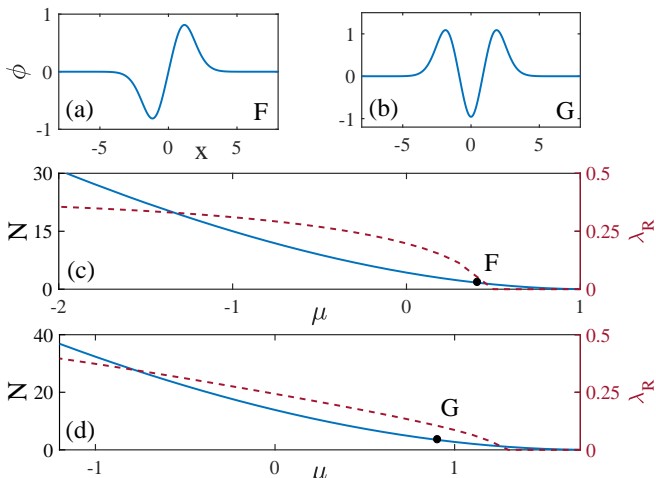

FIG. 2. Profiles, atom number $N$ and its variational approximation one versus chemical potential $\mu$ and harmonic trap's strength $V_0$ for fundamental modes of 1D LHY fluids in the presence of harmonic trap. Profiles of fundamental LHY modes at parameters: (a) $\mu = -1$, $N = 4.65$, and $V_0 = 0.25$; (b) $\mu = -4$, $N = 6.12$, and $V_0 = 0.25$; (c) $\mu = -4$, $N = 5.87$, and $V_0 = 1.2$. (d) Dependence $N(\mu)$ and the variational approximation one (pink dashed line) based on Gaussian trials. (e) Number of atoms ($N$) versus strength of harmonic trap $V_0$ at $\mu = -4$.

FIG. 3. Profiles, atom number $N$ and linear-stability eigenvalues versus chemical potential $\mu$ for higher-order modes of 1D LHY fluids in the presence of harmonic trap of strength $V_0 = 0.25$. Profiles of higher-order LHY modes: (a) dipole soliton at $\mu = 0.4$ and $N = 1.72$; (b) soliton with node $\Bbbk = 2$ at $\mu = 0.9$ and $N = 3.49$. Dependence $N(\mu)$ and maximal real values of the eigenvalues (perturbation growth rate, red dashed line) $\lambda_R$ vs $\mu$ for solitons with different numbers of nodes $\Bbbk$: (c) dipole soliton with node $\Bbbk = 1$; (d) higher-order soliton with node $\Bbbk = 2$.

stable when all real eigenvalues are zero ($\lambda_R \equiv 0$), and unstable otherwise. Dynamical stability of modes $\phi$ is also testified in direct perturbed evolution in Eq. (7).

### III. RESULTS

#### A. LHY fluids without harmonic trap

We first study fundamental modes of 1D LHY fluids in the absence of harmonic trap, their typical profiles and the comparisons with analytical counterparts [Eq. (9)] under different chemical potential $\mu$ are depicted in Figs.1(a) and 1(b), where numerical results and the analytical ones are indistinguishable. It is observed that the amplitude decreases, and the waist (width) expands when increasing $\mu$. The relation between chemical potential $\mu$ and norm $N$ is collected in Fig.1(c), showing that the numerical results can match well with the analytical ones in Eq. (11). The figure also includes the linear-stability results expressed as maximal real values of eigenvalues $\lambda_R$ vs $\mu$, it is observed that the stability property of the fundamental modes follows the trend that they are stable when $\mu$ closes to 0, the weak instability develops when $\mu$ deviates from 0. It is necessary to stress that the stability scenarios in Fig.1(c) can be well verified in direct perturbed evolutions given below.

#### B. LHY fluids with harmonic trap

Including a harmonic trap, we apply variational approximation to construct localized modes. We begin with the Lagrangian of Eq. (8),

$$\mathbb{L} = (\frac{\partial \phi}{\partial x})^2 - 2\mu\phi^2 - \phi^3 + 2V(x)\phi^2. \tag{13}$$

and adopt Gaussian ansatz $\phi = A_0 e^{-\frac{x^2}{2\sigma^2}}$ with amplitude $A_0$ and width $\sigma$, whose norm yields $N = \int_{-\infty}^{+\infty} |\phi(x)|^2 dx = A_0^2 \sigma \sqrt{\pi}$.

With the effective Lagrangian $L_{\text{eff}} = \int_{-\infty}^{\infty} \mathbb{L}dx$, we take Euler-Lagrange equations $\frac{\partial L_{\text{eff}}}{\partial \sigma} = \frac{\partial L_{\text{eff}}}{\partial N} = 0$ to derive the variational equations

$$4V_0\sigma^2 + \sqrt{\frac{2N}{3\sqrt{\pi}\sigma}} - \frac{2}{\sigma^2} = 0, \tag{14}$$

$$\mu = \frac{1}{4\sigma^2} - \frac{3}{4}\sqrt{\frac{2N}{3\sigma\sqrt{\pi}}} + \frac{V_0\sigma^2}{2}. \tag{15}$$

Characteristic shapes of fundamental modes of the 1D LHY fluids in the presence of harmonic trap are displayed in Figs. 2(a), 2(b), and 2(c). The approximate Gaussian solutions based on variational approximation [Eqs. (14) and (15)] match quantitatively with the numerical results constructed from Eq. (8), as seen from the shapes' comparison between

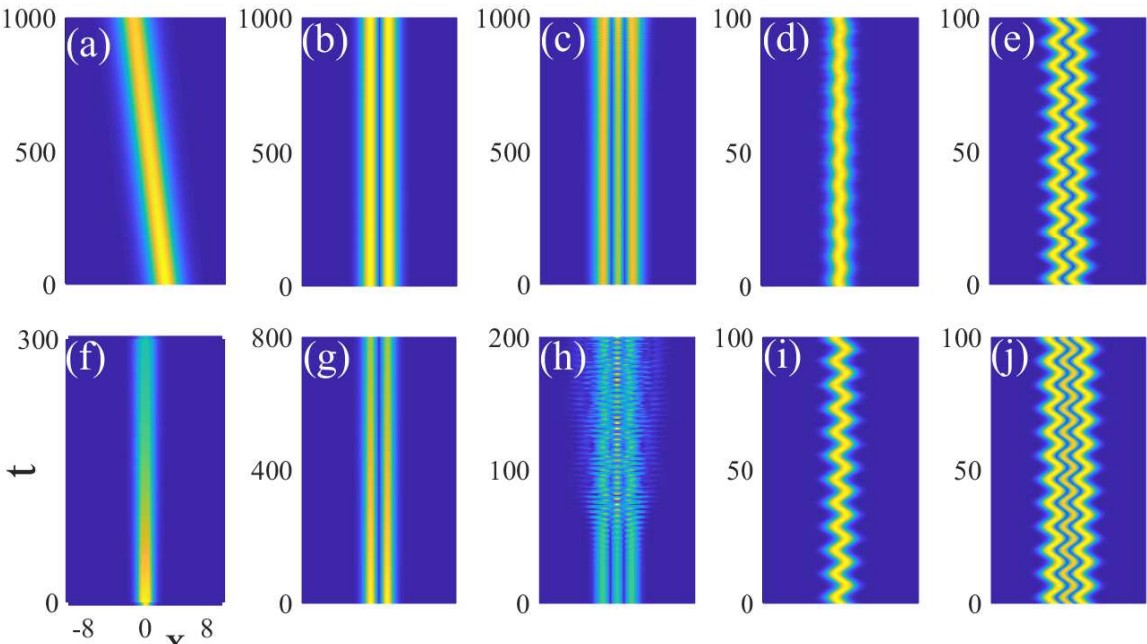

FIG. 4. Stable and unstable evolutions, oscillations of 1D LHY fluids with different numbers of nodes $\Bbbk = 0, 1, 2$. Stable moving (a) and instability (f) of fundamental LHY fluids with $\Bbbk = 0$ without external harmonic trap. Oscillation of LHY fluids with $\Bbbk = 0$ (d, i), 1 (e), 2 (j), and evolutions of fluids with $\Bbbk = 1$ (b, g), 2 (c, h) under the influence of momentum $k_0$ in the presence of harmonic trap of strength $V_0 = 0.25$. Other parameters: (a) $\mu = -0.7$, $k_0 = 0.008$; (b) $\mu = 0.5$, $k_0 = 0$; (c) $\mu = 1.6$, $k_0 = 0$; (d) $\mu = -0.7$, $k_0 = 0.25$; (e) $\mu = -0.1$, $k_0 = 1$; (f) $\mu = -0.7$, $k_0 = 0$; (g) $\mu = -1.8$, $k_0 = 0$; (h) $\mu = -0.5$, $k_0 = 0$; (i) $\mu = -0.7$, $k_0 = 1$; (j) $\mu = -0.9$, $k_0 = 1$.

| $\Bbbk$ | $V_0$ | stability regions ($\mu$) |
|---|---|---|
| 0 | 0 | $-1.2 \leq \mu \leq 0$ |
| 0 | 0.25 | $-1.3 \leq \mu \leq 0.2$ |
| 1 | 0.25 | $0.5 \leq \mu \leq 1$ |
| 2 | 0.25 | $1.3 \leq \mu \leq 1.7$ |

TABLE I. Stability regions (characterized by $\mu$) of all kinds of nonlinear localized modes in 1D LHY fluids with/without harmonic trap.

Figs. 2(a) and 2(b), and from the dependence of number of atoms $N$ on chemical potential $\mu$ in Fig. 2(d). It is also seen from the latter that a small discrepancy between numerical and approximate results appears when $\mu < -1.5$, and above which, in particular, such match is indistinguishable. Noteworthy feature is that all the fundamental modes prepared in harmonic trap are exceptionally stable, verified by direct perturbed simulations and linear-stability analysis. At a defined value of chemical potential ($\mu = -4$), an increase of harmonic trap's strength means to improve the localization ability and therefore more ultracold atoms would be stationed within the trap, explaining the increase relevance of norm $N$ in Fig. 2(e).

All the localized modes reported above are so far confined to fundamental modes, searching higher-order modes of 1D LHY fluids is an interesting issue. Examples of higher-order modes, with different numbers $\Bbbk$ of nodes (zeros), are depicted

in Figs. 3(a) and 3(b). The dipole mode featured by $\Bbbk = 1$ [Fig. 3(a)], and the modes with $\Bbbk = 2$ [like that in Fig. 3(b)], in particular, are a new class of localized modes supported by harmonic trap. To our knowledge, the latter mode has never been found in the setting of BECs with harmonic traps. We further verify that, under the defined strength ($V_0 = 0.25$), the stability regions of both higher-order modes are within a limited physical space in dependency $N(\mu)$ shown in Figs. 3(c) and 3(d), showing are also for their linear-stability analysis.

We would like to point out that the stability regions of all kinds of the nonlinear localized modes of 1D LHY fluids in the presence/absence of a harmonic trap are collected in Table I for easy reading.

### C. Movements, oscillations and collisions of 1D LHY fluids

In the absence of harmonic trap, single localized mode in 1D LHY fluids may be set in motion by adding an initial momentum $k_0$ to stationary wavefunction $\psi(x)$ with a multiplier $e^{ik_0 x}$. Evolution of a stable moving localized mode is depicted in Fig. 4(a); by contrast, an unstable and decaying localized mode like the one in Fig. 4(f) could not maintain good movement since the lost of coherence of a localized mode (soliton). In the presence of harmonic trap, stable higher-order LHY modes with nodes $\Bbbk = 1$ and 2 could keep their shapes during long time evolutions [see Figs. 4(b) and 4(c)], unsta-

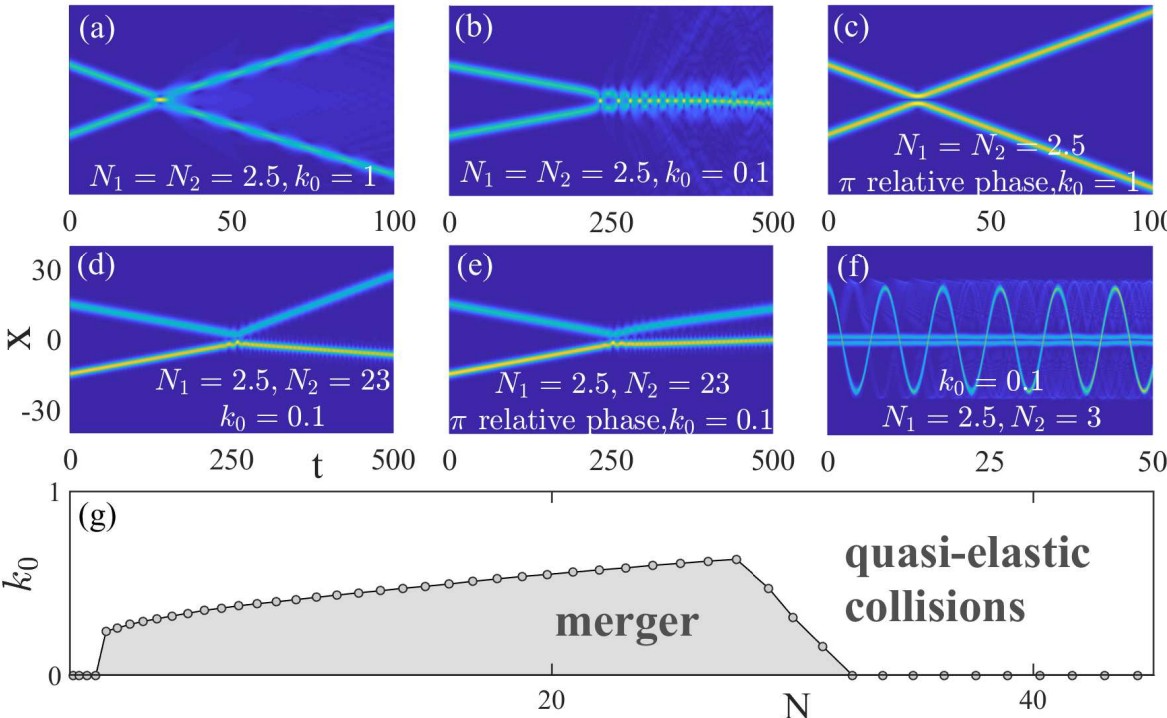

FIG. 5. Collision physics of 1D LHY fluids. (a∼e) Density plot of evolutional dynamics of an interference pattern induced by collision of the two LHY fluids under different incident momentum $k_0$. (f) Collision of a fundamental mode generated from the model without harmonic trap and a dipole mode in harmonic trap. (g) Phase diagram for the collision of the two LHY fluids as a function of incident momentum $k_0$ and the atom number $N$, separating merger and quasi-elastic collisions scenarios associated with those presented in panels (a) and (b).

ble ones decay and oscillate in the evolutions [see Figs. 4(g) and 4(h)]. The multiplication of an initial momentum term will lead to the oscillation of localized modes, displayed in Figs. 4(d) and 4(i) are for fundamental modes with parameters. Regular oscillations apply too to the higher-order modes with nodes $\Bbbk = 1$ and 2, according to the Figs. 4(e) and 4(j).

Another physically relevant signature of the 1D LHY fluids is their quasielastic collisions between the two modes. To implement, initial wave function $\psi(x, t = 0)$ is taken as two counterpropagating fluids

$$\psi(x,0) = e^{i(k_0 x+\varphi)}\phi_1(x+x_0)+e^{-ik_0 x}\phi_2(x-x_0), \quad (16)$$

where $\pm k_0$ are the initial momenta of the two colliding fluids, characterized by their stationary shapes $\phi_1(x)$ and $\phi_2(x)$ produced from Eq. (8), at initial positions $\pm x_0$, and $\varphi$ being the relative phase. The norms of the two fluids are given by $N_1 = \int_{-\infty}^{+\infty}|\phi_1(x)|^2\,dx$ and $N_2 = \int_{-\infty}^{+\infty}|\phi_2(x)|^2\,dx$, respectively. The colliding scenarios are simulated through dynamical equation (7).

We are first interested in the collision between two fluids with the same norm $N_1 = N_2$, such colliding situation is displayed in Figs. 5(a), 5(b), and 5(c). It is observed from the first two panels that an interference pattern forms at the colliding point, leading to the formation of weakly oscillations of two fluids after collision [Fig. 5(a)], and one matter-wave breather [Fig. 5(b)]. While the influence of relative phase on

the collision is significant, according to an example of out-of-phase scattering in Fig. 5(c), where the phase difference $\varphi = \pi$ could induce repulsion between two fluids, resembling those cases in other settings. In terms of collision of two fluids with unequal norms, as shown in Figs. 5(d) and 5(e) [$\varphi = \pi$], quasielastic scattering happens since the two fluids evolve into oscillatory modes, depending not on their relative phase.

It is a challenging issue to realize the collision of two LHY fluids in the presence of harmonic trap. We devise a scheme to realize the collision between the two fluids with unequal norms, with one fluid as fundamental mode generated without the trap (or by switching off the trap), then use it to collide with another mode formed inside the trap. As an example in Fig. 5(f), a fundamental mode is collided with a dipole mode, the fundamental LHY fluid evolves into an oscillating mode within the harmonic trap, and exchanges energy with stationary dipole mode in every collision.

In Fig. 5(g) , we have summed the collision scenarios of two LHY fluids with identical atom number $N_1 = N_2 = N$ in the absence of trap, expressed as a function of incident momentum $k_0$ and $N$, emphasizing the separation of merger and quasi-elastic collisions. When the two LHY fluids with unequal norms $N_1 \neq N_2$ are collide, quasi-elastic scattering is always the feature during the collision, such effect is not included in Fig. 5(g) but has been displayed in Figs. 5(d) and 5(e).

## IV. CONCLUSION AND DISCUSSION

We have investigated, analytically and numerically, the localized modes of 1D purely LHY fluids governed by quantum fluctuations in a mixture of BECs in the framework of Gross-Pitaevskii equation with attractive quadratic nonlinearity which, in essence, is different from the 3D LHY fluids with repulsive quadruple nonlinearity [49]. We also obtained approximate solutions for fundamental mode based on variational approximation and numerically constructed higher-order localized modes with nodes $\Bbbk = 1$ and 2 for quantum fluids in a harmonic trap. The properties and dynamics of all the localized modes were testified by linear-stability analysis and direct perturbed simulations. Further, depending on the initial momenta, movements and oscillations of single localized mode, and quasielastic collisions between two modes,

were reported in time evolution. The predicted solutions are accessible in quantum-gas experiments, providing deep insights into low-dimensional LHY fluids. We notice from [49] and [50] that the monopole breathing mode is the signature of LHY fluids, which deserves to be revealed in future studies in low-dimensional settings.

### Acknowledgements

This work was supported by the National Natural Science Foundation of China (NSFC) (No.12074423) and Young Scholar of Chinese Academy of Sciences in western China (No.XAB2021YN18).

### Conflict of Interest

The authors declare no conflicts of interest.

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
