# Peer review of "One-dimensional purely Lee-Huang-Yang fluids dominated by quantum fluctuations in two-component Bose-Einstein condensates"

_SciPost Physics_

## Round 1 · Referee Report · Anonymous · 2021-9-14

Strengths

The revised manuscript adequately addresses all the points from the original review. Thus, strengths remain the same as before, to which an essentyially clearer presentation is added.

Weaknesses

Weakness mentioned in the original review have been fixed in the revised manuscript.

Report

I think the revised manuscript meets all acceptance conditions.

---

## Round 1 · Author Response

We thank the two anonymous reviewers for reviewing our work and providing very useful comments which help to improve the quality of this paper.

---

## Round 1 · List of Changes

Reply to the First Reviewer—scipost_202107_00032v1

Point 1: In particular, I surmise that there are typos in Eqs. (1) and (3), as it seems strange that the second terms in those expressions do not include any density of the condensate.
Reply to this point: We think this is a misunderstanding. Actually, there are not typos in Eqs. (1) and (3), the density of the condensate is included in parameter “$c_{\pm}$” which is given in Eq. (2).

Point 2: By the way, what is meant by the "momentum integration" of Eq. (1)?
Reply to this point: The meaning of “momentum integration” is to integrate in momentum space, just like the ones usually do in real space to reduce the 3D space to the quasi-one-dimensional case in tackling Bose-Einstein condensate. Such term is not coined by us, and has been used in a seminal work by Petrov and Astrakharchik in Ref. [28], which we have cited to describe such term in the first version.

Point 3: Particularly, the statement that Eq. (9) is a completely new solution seems disputable, as it is a special case of a more general solution (16) for the "droplets", reported in Ref. [28].
Reply to this point: This comment is relevant. We have revised the corresponding expression and cited that paper. Our new description is attached here:” While based on the previous work for 1D quantum droplets in quadratic-cubic model in Ref. [28], discarding the cubic term in where can lead to an exact solution for our model Eq. (8) in the absence of harmonic trap”.

Point 4: Another inaccuracy is the reference to Fig. 5(b) as the formation of "two matter-wave breathers", while in reality one observes merger of colliding droplets into a single breather in this figure.
Reply to this point: This comment is relevant too. We have replaced the “two matter-wave breathers” by “one matter-wave breather”.

Point 5: Lastly, it is recommended to present results for collisions between the "droplets" in a more systematic form. In particular, it is relevant to find a critical collision velocity separating quasi-elastic collisions and the merger.
Reply to this point: This suggestion is very relevant. We have calculated all the possible cases and produced a new figure, which is the current Fig. 6 in this revised version; and in the section before conclusion, we add a paragraph to describe this figure, which is attached as follows:
“In Fig. \ref{fig6}, we have summed the collision scenarios of two LHY fluids with identical atom number $N_1=N_2=N$ in the absence of trap, expressed as a function of incident momentum $k_0$ and $N$, emphasizing the separation of merger and quasi-elastic collisions.”

Reply to the Second Reviewer
Point 1: The analytic droplet solution has been obtained in Ref.[28] in the general case, including the point of vanishing mean field.
Reply to this point: This comment is relevant. We have revised the corresponding expression and cited that paper. Our new description is attached here:” While based on the previous work for 1D quantum droplets in quadratic-cubic model in Ref. [28], discarding the cubic term in where can lead to an exact solution for our model Eq. (8) in the absence of harmonic trap”.

Point 2: The stability analysis (Bogoliubov modes) of the ground-state droplet solution in free space has been performed by Tylutki et al., Phys. Rev. A 101, 051601 (2020). The droplet is found stable. I thus do not understand the authors' conclusion on "very limited stability interval for fundamental modes" in this case. Accordingly, I also do not understand Fig.4d. The validity is questionable.
Reply to this point: Actually, our model is a purely quadratic attractive nonlinear model, very different from the quantum droplets with competing nonlinear models like the 1D case with attractive quadratic and repulsive cubic nonlinear terms (as in the mentioned reference, which has been cited as Ref. [44]), thus the stability property is different. In quadratic nonlinear model, the nonlinear localization effect cannot be simply duplicated from the cubic (Kerr) model either. To clarify this, our conclusion and the Fig.4d are appropriate.

Point 3: Figures 4b, 4c, 4e, and 4f seem to show trivial center-of-mass dipole oscillations in the harmonic trap. I see no monopole mode there. I thus doubt that this figure is useful.
Reply to this point: For the cases in Figs. 4b, 4c, 4e, and 4f, the oscillation was induced by initial momentum $k_0$, making the fundamental modes and higher-order ones out of equilibrium (but not evolve themselves into new possible localized modes), while they are still trapped by harmonic trap; such effect has been explained well in the original draft. Such effect can in a certain imply the robustness of the finding localized modes, and is thus useful, we keep it in this revised version too.
Concerning the second point about the monopole mode, we thank this reviewer for reading our draft very carefully and for stressing this misunderstanding point made by us. In this revised version, we improve our understanding and expressions related to the above subfigures. Since the monopole oscillations and the associated monopole modes are described by some very professional ways and which are out of our specialty, and we thus let them open; to this, we rewritten such expressions at the last sentence before talking about collision of two LHY fluids, where we previously stated that “We notice that the oscillations of 1D LHY fluids presented here, briefly speaking, resemble the monopole oscillations of a 3D LHY fluid confined in a spherical potential observed recently in a mixture of BECs Refs. [69] and [70].”, such expressions are deleted right now, and at the end of the Conclusion part, we have added a sentence to clarify this, which is stated as “We notice from Refs. [69] and [70] that the monopole breathing mode is the signature of LHY fluids, which deserves to be revealed in future studies in low-dimensional settings.”

Point 4: There is a significant overlap between this manuscript and the work of Astrakharchik and Malomed, Phys. Rev. A 98, 013631 (2018) who studied collisions between droplets in a slightly more systematic manner.
Reply to this point: The collisions between two LHY fluids indeed resemble some similarities as those for 1D quantum droplets in Astrakharchik and Malomed, Phys. Rev. A 98, 013631 (2018), which is cited as Ref. [43]. A flat-top profile is a signature of the quantum droplets, provided by the competing attractive quadratic and repulsive cubic nonlinear terms. To provide dynamical properties of the finding fluids, such collisions could tell the rich dynamics during colliding evolution, while the physical model for LHY fluids considered here keeps only the attractive quadratic term, which is radically different from the 1D quantum droplets presented in Ref. [43].

Point 5: There are many inconsistencies. Some examples are the absence of "yellow dashed line" in Fig.2d, no "shapes' comparison" in Figs.2a and 2b, no labels a, b, c, and d in Fig. 3, etc. The authors should have carefully read their draft before submitting.
Reply to this point: We thank this reviewer for reading our draft very carefully and pointing out inconsistent expressions. In this revised version, "yellow dashed line" in Fig. 2d has been corrected as “pink dashed line”; "shapes' comparison in Figs. 2a and 2b" were replaced by "shapes' comparison between Figs. 2a and 2b"; and the labels a, b, c, and d have been added to Fig. 3.

Point 6: The potential impact of the paper could be improved if the authors better explain physical motivation for studying higher-order modes. It would also help if they discuss stability of these modes in a more systematic manner. Are there stability thresholds for excited modes in Fig. 3? If yes, what is their physical meaning?
Reply to this point: The higher-order modes could tell more details about the robustness and rich dynamical properties of the underlying physical model. We have added four subfigures, denoted as Figs. 4b, 4c, 4g and 4h, to show the stability and instability of these higher-order modes in the current version. The stability thresholds can be seen from the maximal real values of the eigenvalues (perturbation growth rate, red dashed line) λR in Fig. 3, the higher-order modes are stable only when λR=0. The stability thresholds tell us the excitation of stable higher-order modes can be possible only within some certain value of chemical potential, and such condition changes with the variation of harmonic trap.

You are currently on this page

Resubmission 2109.05515v1 on 14 September 2021

---

## Round 2 · Referee Report · Anonymous · 2021-10-13

Report

I am not convinced by the author's response. They want to publish the article at any cost. I am pretty sure that there is a problem with validity and they refuse to check it seriously. I do not buy the argument that the point of vanishing mean field is special in terms of stability.

---

## Round 2 · Referee Report · Anonymous · 2021-10-14

Strengths

1. Figure 6 is interesting and provides some physical intuition for the collision physics.

Weaknesses

1. No clear discussion of main advances over previous published works on the subject where the vanishing mean field limit can be directly ascertained by sending a parameter to zero in equations.
2. Limited physical motivation provided for the reported lack of stability for the fundamental modes.
3. Discovery of an exact solution for the fundamental mode of LHY quantum fluids is still reported in the abstract and conclusions.

Report

While the authors have significantly toned down claims in earlier versions of a new exact solution for the profile of Lee-Huang-Yang droplets, it still appears in the abstract and conclusion, i.e. "An exact solution is found for fundamental LHY fluids" and "An exact solution for fundamental mode of LHY fluids without any external trap was derived". As pointed out by the second referee, Eq. (9) is a parametric limit of a previously derived expression in Ref. [28] (Eq. (16)).

In my opinion, the authors have also failed to satisfactorily address the concerns of the second referee around the stability of the fundamental mode. The authors claim that "the stability property of 1D LHY fluids is very different to that of 1D quantum droplets" but don't back this up in the manuscript with any specific reasoning. For example, can it be shown how this stability is recovered perturbatively in the prefactor of the quartic term? I am also somewhat concerned by the two kinks in the dashed line in Figure 1. What is the origin of these? Their presence could indicate an issue or instability in the solution.

I also have similar concerns to the previous referees on readability. While the authors have made improvements, there are still residual issues, for example:

1. what is the yellow dashed line in Figure 1?
2. Below Eq. (13) there is a sentence on :"shape's comparison" between numerical and approximate solutions. I don't see such a comparison in the figure.
3. The inequality \mu > -1.5 seems flipped to me, I see a discrepancy for smaller values of \mu. This sentence should be clarified.
4. The organization of panels in Figure 4 is very confusing. Perhaps an additional labelling scheme for trap vs. no trap is needed.

  • validity: low
  • significance: low
  • originality: low
  • clarity: low
  • formatting: reasonable
  • grammar: below threshold

---

## Round 2 · Author Response

We thank you again for reviewing our work and providing very useful comments which help to improve the quality of this paper.

---

## Round 2 · List of Changes

Point 1: "We find a new exact solution as bright solitons..."
"...exact solution has not been reported before for purely quadratic nonlinear model (without harmonic trap)."

Reply: "We find a new exact solution as bright solitons..." have been revisited as “We find an exact solution as bright solitons (which can be derived from the 1D quantum droplets by ignoring the mean-field term [28, 43, 44])”.
This statement "...exact solution has not been reported before for purely quadratic nonlinear model (without harmonic trap)." has been deleted.

Point 2: The relation between the chemical potential and the atom number is also known analytically. The shapes of LHY solitons for different values of the chemical potential are obviously self-similar. Checking these results numerically (Fig.1) does not merit publication.

Reply: We think it is necessary to show the shapes of 1D LHY fluids, since they are different from the previous obtained exact solutions for 1D quantum droplets in Ref. [28, 43, 44] where the mean-field cubic term keeps always, while here in our model we just consider the LHY term.

Point 3: I repeat that I do not understand the red dashed line in Fig.1 and the statement on the "very limited stability interval for fundamental modes". There is a problem either with validity or with interpretation. The authors basically argue that the soliton can be unstable in free space and that there is a stability threshold (since they speak about a "stability interval"). That the soliton is stable has been shown in Ref.[44] and the absence of a threshold follows from the property of self-similarity. Figure 4f (in the new version) requires more explanations. In what sense does it show that the soliton is unstable?

Reply: To facilitate the reading, the statement on the "very limited stability interval for fundamental modes" concerning the red dashed line in Fig.1 has been revisited as “where includes also the linear-stability results expressed as maximal real values of eigenvalues $\lambda_R$ vs $\mu$ , it is seen from the latter that the fundamental modes have a very limited stability interval, recalling that the solutions are stable only when $\lambda_R=0$.”
We stress once again that the stability property of 1D LHY fluids is very different to that of 1D quantum droplets. In evaluating the stability property of LHY fluids, the stability is first checked in linear stability analysis method and further confirmed in direct perturbed simulations, and both methods can match well; and we find that the linear stability analysis results for the fundamental modes have a very limited interval for $\lambda_R=0$. While for the 1D quantum droplets, the focusing and defocusing competing nonlinearities could help to improve the stability of their modes.
Concerning Figure 4f , we state that “an unstable and decaying localized mode (the corresponding number of atoms reduces) like the one in Fig. 4(f) could not maintain good movement since the loss of coherence of a localized mode (soliton).”, thus the soliton is unstable since the total number of atoms diminishes during the evolution.

---

## Editorial Decision

editor-in-charge_assigned